

# ACCOUNTING FOR METEOROLOGICAL EFFECTS IN THE DETECTOR OF THE CHARGED COMPONENT OF COSMIC RAYS

Maxim Philippov[1], Vladimir Makhmutov[1], Galina Bazilevskaya[1], Fedor Zagumennov[2,3], Vladimir Fomenko[2], Yuri Stozhkov[1], Andrey Orlov[1]

[1]P. N. Lebedev Physical Institute of the Russian Academy of Sciences, Moscow, Russian Federation

[2]Federal State Budgetary Institution «Central Aerological Observatory», Dolgoprudny, Russian Federation

[3]Plekhanov Russian University of Economics, Moscow, Russian Federation

*Correspondence to:* Maxim Philippov (mfilippov@frtk.ru)

**Abstract.** In this paper, we discuss the influence of meteorological effects on the data of the ground installation CARPET, which is a detector of the charged component of secondary cosmic rays (CRs). This device is designed in the P.N. Lebedev Physical Institute (LPI, Moscow, Russia) and installed at the Dolgoprudny scientific station (Dolgoprudny, Moscow region, S55.56 °, 

W37.3 °; R$c$ = 2.12 GV) in 2017. Based on the data obtained in 2019–2020, the barometric and temperature coefficients for the CARPET installation were determined. The barometric coefficient was calculated from the data of the barometric pressure sensor included in the installation. To determine the temperature effect, we used the data of upper-air sounding of the atmosphere obtained by the Federal State Budgetary Institution «Central Aerological Observatory» (CAO), 

also located in Dolgoprudny.

*Keywords*: cosmic rays; CARPET detector; meteorological effects

## 1.    Introduction

The CARPET installation is designed for permanent monitoring of charged component of secondary CRs flux at the ground level. It allows analysis of secondary CRs fluxes variations, 

caused by geomagnetic and solar activity on the processes affecting the behavior of cosmic rays in near-Earth space and Earth's atmosphere (Makhmutov et al., 2013, 2015).

The basis of the CARPET installation (Fig. 1) is the STS-6 gas-discharge Geiger – Müller counters, combined in 12 detector blocks of 10 counters each. The detector block consists of two layers: 5 upper and 5 lower counters, separated with an aluminum absorber (filter) 7 mm thick. Experimental 

data are recorded using three channels with a time resolution of 1 ms. The first channel (UP) corresponds to the integral count rate of charged particles passing through the top layer of 60 counters. The second channel (LOW) corresponds to the integral count of charged particles passing



through the bottom layer of 60 counters. Particles simultaneously registered by any of the upper and lower counters, i.e., passed through the filter are registered in the coincidence channel - TEL.

In addition, there is a channel of auxiliary information ("telemetry"), which consist of the data on pressure, temperature and supply voltages.

The CARPET installation detects particles of the following energies: in the UP and the LOW channels there are electrons and positrons with energies E> 200 keV, protons with E> 5 MeV, muons with E> 1.5 MeV, and photons with E> 20 keV (efficiency <1%). The TEL coincidence

channel registers more energetic particles: electrons with energies E> 5 MeV, protons with E> 30 MeV, and muons with E> 15.5 MeV. Detailed information on the principles of CARPET operation was given previously (Philippov et al., 2020a).

Nowdays there is an international network of the CARPET installations: first module was launched in 2006 (De Mendonca et al., 2011, 2013; Mizin et al., 2011) at CASLEO (San Juan, Argentina,

S31.47°, W69.17°, R*c* = 9.8GV), two modules were launched (Maghrabi et al., 2020) in 2015 at KACST (King Abdulaziz City for Science and Technology, Saudi Arabia, Riyadh, S24.39°, E46.42°; R*c* = 14.4GV). In 2015 and 2016 at L.N. Gumilyov Eurasian National University (Nur-Sultan, Republic of Kazakhstan, S51.10°, W71.26°; R*c* = 2.9 GV), the first and second modules of the CARPET installation were launched (Philippov et al., 2020b; Tulekov et al., 2020).

This paper investigates the influence of meteorological conditions on the data of the installation, which has been operating since 2017 at the Dolgoprudny Scientific Station of the Lebedev Physical Institute RAS.

## 2. Instrumentation and data analysis

### 2.1 Barometric effect

Ground-based CARPET installations detect secondary charged particles, mainly muons, generated in the interaction of primary CRs with nuclei in the atmosphere. Muons are not nuclear-active particles and lose energy for the excitation and ionization of air atoms; therefore, it is necessary to take into account the barometric and temperature effects (Dorman, 1972, 2004, 2006).

The barometric effect can be determined through variations in atmospheric pressure at the level of

CRs registration (equation 1):

$$\left(\frac{\Delta N}{N_0}\right)_P \cong \beta \Delta P, \tag{1}$$

where

$\left(\frac{\Delta N}{N_0}\right)_P$ – relative variation of the count rate of the CARPET installation;





$\Delta N = N - N_0$;

$\Delta P = P - P_0$;

$N_0$ – average (standard) count rate [pulses/h] for the period of measurements;

$N$ – current count rate [pulses/h];

$P_0$ – average (standard) ground atmospheric pressure [hPa] for the period of measurements;

$P$ – current atmospheric pressure [hPa].

According to the data for 2019, hourly averaged average count rate and atmospheric pressure for

the CARPET-MOSCOW installation $N_0 = 53667$ pulses/h, $\sigma_N = 2187$ pulses/h; $P_0 = 988.7$ hPa, $\sigma_P$ = 9.8 hPa.

For calculating the barometric coefficient $\beta$, it is necessary to determine the linear relationship between $\frac{\Delta N}{N_0}$ and $\Delta P$ (Fig. 2). Barometric coefficient $\beta$ for the CARPET-MOSCOW installation is determined on the data of June 2019 (During this period there were no significant geomagnetic

and solar disturbances): $\beta = -0.1861 \pm 0.0025$%/hPa; coefficient of determination $R^2 = 0.8975$. Using Eq. (1), we obtain pressure-corrected data:

$$N_{PC} = N - \beta N_0 \Delta P, \qquad\qquad\qquad\qquad (2)$$

where

$N_{PC}$ – pressure corrected count rate [impulses/h] of the CARPET installation.

To estimate primary CRs variations, we use pressure corrected data of the Moscow neutron

monitor (http://cr0.izmiran.ru/mosc/). Average count rate according to the data of 2019: $\overline{N_{nm}} = $ 9699 pulses/min; $\sigma_{nm} = 66$ pulses/min.

Fig. 3 shows neutron monitor count rate variations on the data of 2019. Black horizontal line is average count rate [pulses/min]. Upper horizontal data series is standard deviation from the average count rate for each month. The relative magnitude of the effect determined by the

variations in primary CRs over a given period of time can be estimated by the ratio $\sigma_{nm}/\overline{N_{nm}} = $ 0.007 (0,7%).

Magnitude of the barometric effect of the CARPET-MOSCOW can be estimated as $\beta \sigma_P = 0.018$ (1.8%), which is more than 2 times higher than variations of primary CRs. Therefore, the barometric effect is significant for the CARPET installations and must be taken into account

in the further data processing.

## 2.2    Temperature effect

The muon component of secondary CRs is characterized by a significant temperature effect (Yanke, et al., 2011), to eliminate which it is necessary to carry out upper-air sounding near the instrument. The temperature effect has two components: negative and positive. The negative





temperature effect is associated with a decrease in muon fluxes during heating and expansion of
the atmosphere. The positive temperature effect is associated with the appearance of additional
muons, as a result of an increase in the rate of decays of charged pions (Dorman, 1972, 2004, 2006;
Yanke et al., 2011).

To estimate the temperature effect, we used data of the TEL channel of the CARPET – MOSCOW

installation for 2019–2020. The altitude profiles of temperature and pressure were determined
from the experimental data of the Central Aerological Observatory (CAO; Dolgoprudny).

The temperature effect was determined in two ways: based on the effective generation level
method and the integral method (Dmitrieva et al., 2013; Ganeva et al., 2013; Zazyan et al., 2015).

### 2.2.1  Effective generation level method

To eliminate the barometric effect, original data (Fig. 4a) were processed according to Equation 1
(Fig. 4b). The barometric correction mainly compensates for the daily variations in the count rate.
The effective generation rate method is based on the assumption that muons are mainly generated
at a certain isobaric level, which is 100 hPa (Dmitrieva et al., 2013). The height $H$ of this level
depends on the atmospheric temperature. The deviation of the count rate of the installation,

therefore, depends on the change in the height of the generation level $\Delta H$ and the change in the
temperature of this layer of air:

$$\left(\frac{\Delta N}{N_0}\right)_T = \alpha_H \Delta H + \alpha_T \Delta T \qquad (3)$$

where

$\left(\frac{\Delta N}{N_0}\right)_T$ – count rate relative variations of the CARPET installation;

$\Delta H$ – absolute deviation of the effective generation level [km];

$\alpha_H$ – negative temperature coefficient [%/km];

$\Delta T$ – absolute temperature deviation at the level of effective generation [°C];

$\alpha_T$ – positive temperature coefficient [%/°C].

Upper-air meteorological sondes are launched twice a day, at 11:30 and 23:30 UTC. The picture
of a typical MRZ-3AK1 sonde is presented in Fig. 5. Flights last, on average, about 1.5 hours,

therefore, from the available data of the CARPET-MOSCOW installation were made samples of
hourly data from 12:00 to 13:00 UTC and 00:00 to 01:00 UTC.

To calculate the contribution of the negative component of the temperature effect, we define the
linear relationship between $\frac{\Delta N}{N_0}$ and $\Delta H$ (Fig. 6),

where

$\Delta N = N_{PC} - N_0$;





$\Delta H = H - H_0$;

$H_0$ – average (standard) height of the level of effective generation [km] for 2019–2020;

$H$ – current height of the level of effective generation [km].

For the CARPET-MOSCOW installation: $H_0 = 16.1$ km, $\sigma_H = 0.3$ km. Using the least squares
method, we define the approximating line, the slope of which is equal to $\alpha_H$.

$\alpha_H = -4.00684 \pm 0.0652\%/$km; coefficient of determination $R^2 = 0.8191$.

Corrected data series (Fig. 4c) is calculated by the equation:

$$N_{HPC} = N_{PC} - \alpha_H N_0 \Delta H, \qquad\qquad (4)$$

where

$N_{HPC}$ – count rate [pulses/h] of the CARPET installation with negative temperature effect
correction.

To calculate the contribution of the positive component of the temperature effect, we define the
linear dependence between $\frac{\Delta N}{N_0}$ and $\Delta T$ (Fig. 7),

where

$\Delta N = N_{HPC} - N_0$;

$\Delta T = T - T_0$;

$T_0$ – average (standard) temperature at the level of effective generation [°C] for 2019-2020
according to CAO measurements;

$T$ – current temperature at the level of effective generation [°C].

$T_0 = -56.9$ °C, $\sigma_T = 6.0$°C.

Using the least squares method, we define the approximating line, which slope is $\alpha_T$.

$\alpha_T = 0.0080 \pm 0,0038\%/$°C; coefficient of determination $R^2 = 0,0049$.

As seen in Fig. 7, there is a slight positive temperature effect. Corrected data series is calculated
by the equation (Fig. 4d):

$$N_{THPC} = N_{HPC} - \alpha_T N_0 \Delta T, \qquad\qquad (5)$$

where

$N_{THPC}$ – count rate [pulses/h] of the CARPET installation with positive temperature effect
correction.

### 2.2.2 Integral method

Consider the integral method for determining the temperature effect:

$$\left(\frac{\Delta N}{N_0}\right)_T = \int_0^P \alpha(x)\Delta T(x)dx \qquad\qquad (6)$$

where



$P$ – atmospheric pressure at the point of determination of the temperature effect;

$\alpha(x)$ – temperature coefficient density;

$\Delta T(x)$ – temperature deviation from the average value in the air layer corresponding to the pressure
from $x$ to $x+dx$.

There are 16 isobaric surfaces commonly accepted while analyzing upper-air atmospheric effects:
1000, 925, 850, 700, 500, 400, 300, 250, 200, 150, 100, 70, 50, 30, 20, and 10 hPa. They are also
used in observations by CAO. It was decided to exclude the surface of 10 hPa from the
calculations, since for the time period 2019 - 2020 there are only 148 measurements for this
isobaric surface pressure level.

Represent equation 6 as a sum:

$$\left(\frac{\Delta N}{N_0}\right)_T = \sum_P \alpha(P)\Delta T(P) \tag{7}$$

where

$\alpha(P)$ – temperature coefficient for a given isobaric surface [%/°C];

$\Delta T(P)$ – deviation of temperature from the average value for a given isobaric surface [°C].

Starting from the first isobaric surface (20 hPa), we will determine the dependence between $\frac{\Delta N}{N}$
and $\Delta T$. The corrected data for the first surface is then used to determine the temperature
coefficient for the next surface, and so on:

$$N_{i+1} = N_i(1 - \alpha_{i+1}\,\Delta T_{i+1}), \tag{7}$$

where

$\alpha(P)$ – temperature coefficient of the isobaric surface $i+1$ [%/°C];

$\Delta T(P)$ – temperature deviation from the average value for the isobaric surface $i+1$ [°C];

$N_i$ – count rate of the CARPET-MOSCOW, with temperature correction along the isobaric surface
$i$;

$N_{i+1}$ – count rate of the CARPET-MOSCOW, with temperature correction along the isobaric
surface $i+1$;

The results are shown in Table 1: the first column is the atmospheric pressure on the given surface,
the second column is the average temperature according to the data for 2019 - 2020, the third
column is the standard deviation of the temperature, the fourth column is the temperature





coefficient for the given isobaric surface, the fifth column is number of measurements. In fig. 4e shown the count rate of the CARPET-MOSCOW installation, corrected with integral method, according to the data for 2019 - 2020.

### 3. Conclusion

This paper describes the CARPET installation, designed for detecting the charged component of secondary CRs. The barometric coefficient was determined using the built-in pressure sensor. The temperature coefficient was determined by two methods using the data of the upper-air sounding. The results obtained by the effective generation method and the integral method correlate with each other. In this connection, it is more optimal to use the method of the effective generation

level, since it does not require a complete temperature profile. Also, for the CARPET-MOSCOW installation, it is possible to use only the negative component of the temperature effect, since variations of the count rate have good ($R^2 = 0.8191$) correlation with $\Delta H$.

*Data availability*. Data related to this article are available upon request to the corresponding authors.

*CRediT author statement*

**M. Philippov:** Conceptualization, Methodology, Software, Electronics, Data curation, Writing-Original draft preparation,

**V. Makhmutov:** Conceptualization, Methodology, Data curation, Writing- Original draft preparation,

**G. Bazilevskaya:** Conceptualization, Writing- Original draft preparation,

**F. Zagumennov:** Data curation, Original draft preparation,

**V. Fomenko:** Data curation,

**Yu. Stozhkov:** Conceptualization,

**A. Orlov:** Data curation.

### 4. Acknowledgments

The authors express their gratitude to the Neutron Monitor Database (NMDB) team (www01.nmdb.eu) for the data from the ground network of neutron monitors.



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





**Figures**

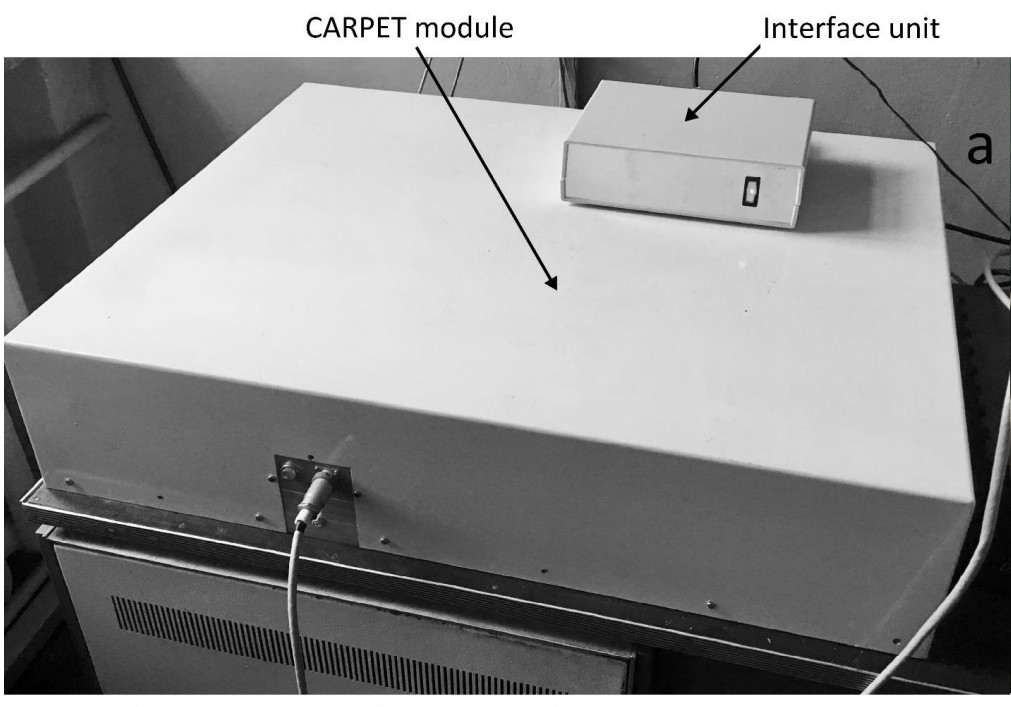

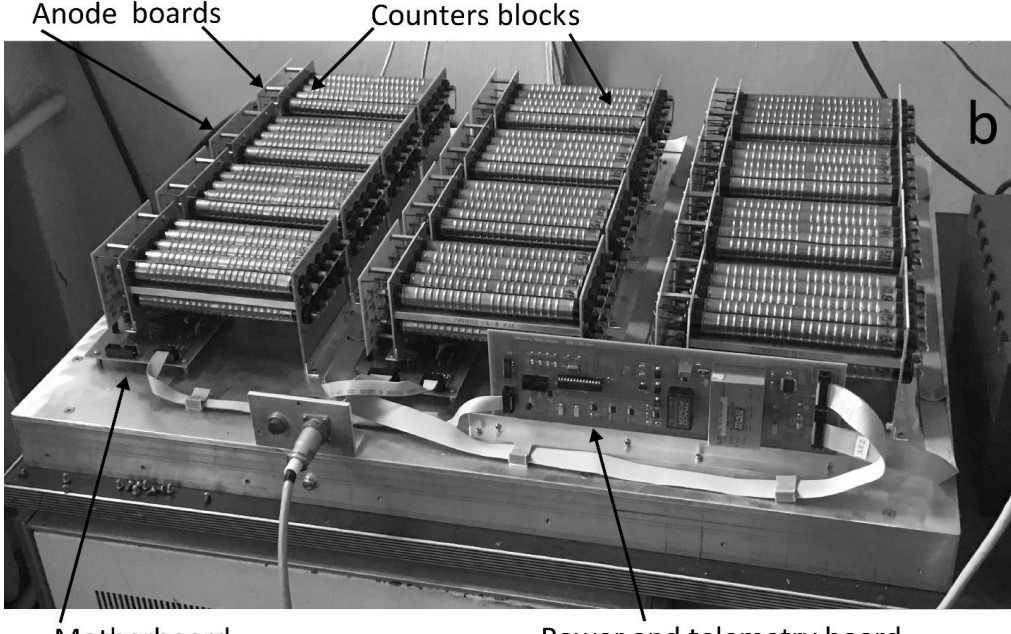


**Fig. 1.** CARPET-MOSCOW installation and its components. On panel $a$ − CARPET module with cover and on panel $b$ −CARPET module without cover.





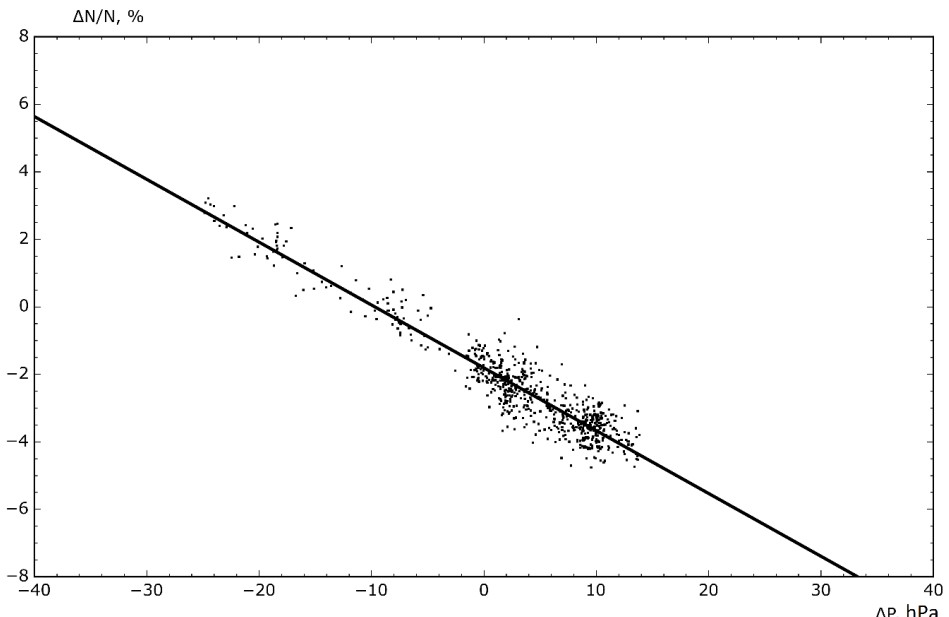

**Fig. 2.** Relationship between $\frac{\Delta N}{N_0}$ and $\Delta P$ for the CARPET-MOSCOW installation determined on the data of June 2019

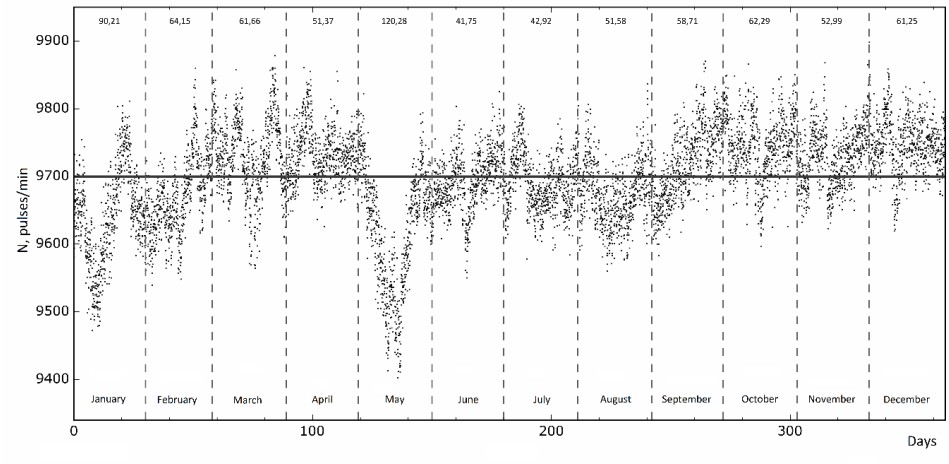

**Fig. 3.** Count rate variations of the Moscow neutron monitor for the period of 2019. Horizontal line – average count rate.





**Fig. 4.** Count rate variations of the CARPET-MOSCOW installation: *a* – uncorrected data, *b* – pressure corrected data, *c* - pressure and temperature (negative effect) corrected data, *d* – pressure



and temperature (negative and positive effect) corrected data, $e$ – pressure and temperature

(integral method) corrected data.

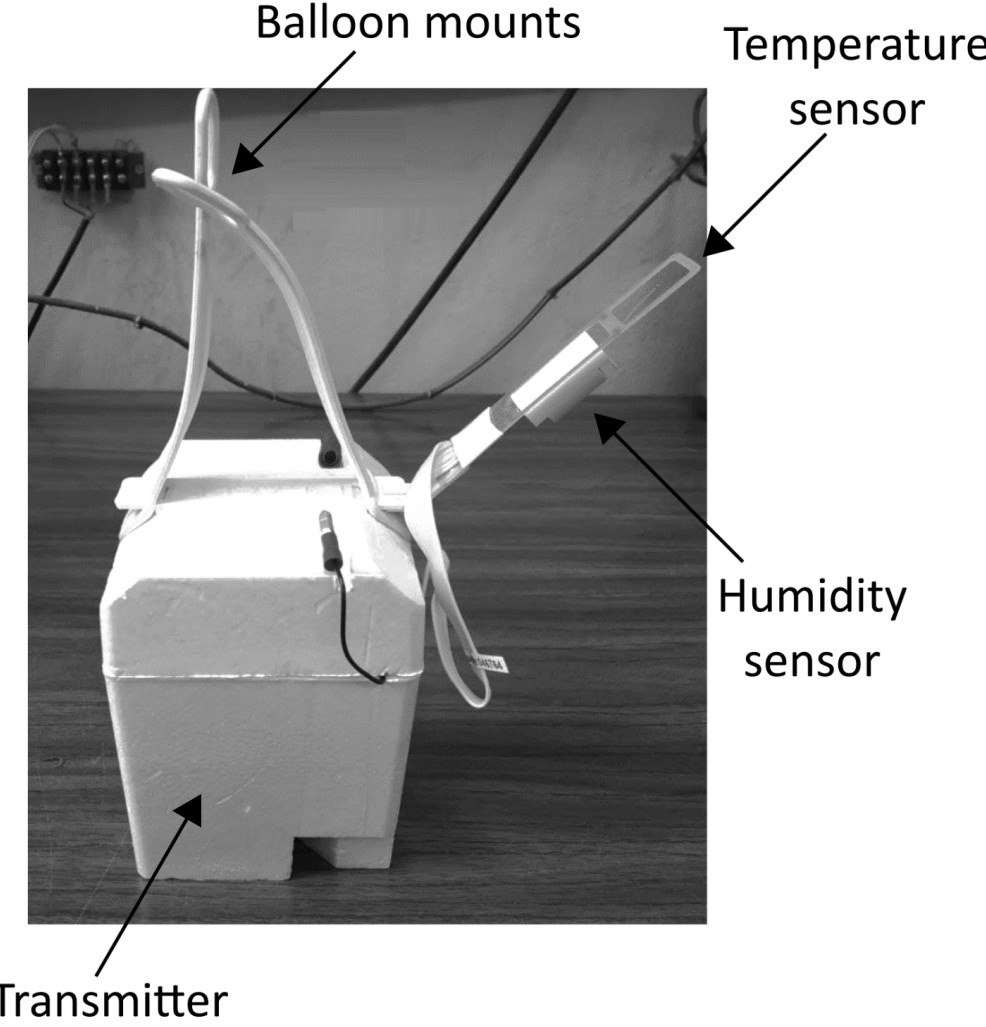

**Fig. 5.** Upper air sonde MRZ-3AK1 (CAO; Dolgoprudny)



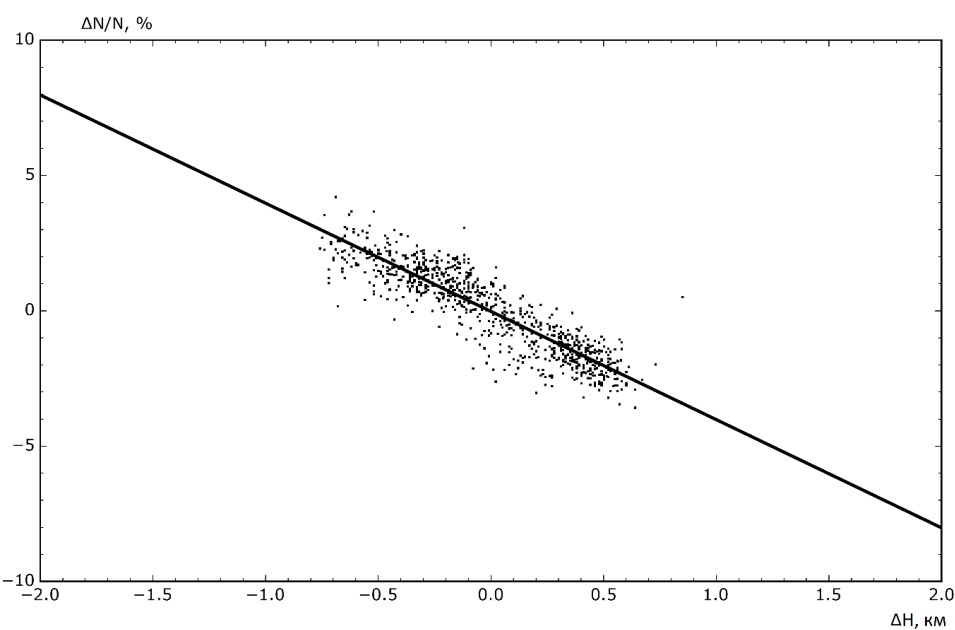


**Fig. 6.** Relationship between $\frac{\Delta N}{N_0}$ and $\Delta H$ (negative temperature effect) for the CARPET-MOSCOW

installation determined on the data of 2019-2020

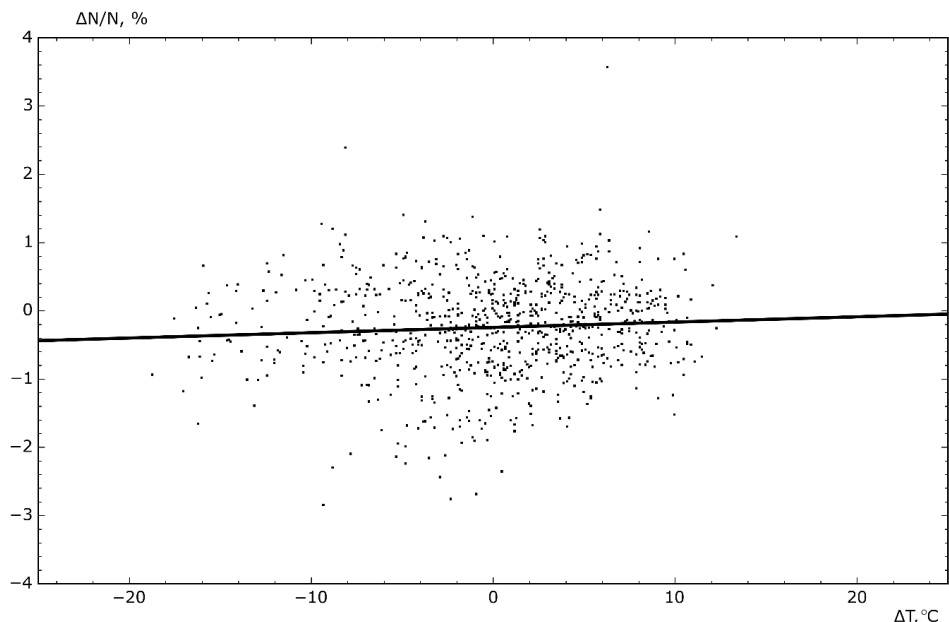

**Fig. 7.** Relationship between $\frac{\Delta N}{N_0}$ and $\Delta T$ (positive temperature effect) for the CARPET-MOSCOW

installation determined on the data of 2019-2020



**Tables**

| $P$, hPa | $\overline{T}$, °C | $\sigma_T$,°C | $\alpha$, %/°C | $n$ |
|---|---|---|---|---|
| 20 | -57,13 | 11,30 | -0,0909±0,0041 | 670 |
| 30 | -59,00 | 9,04 | -0,0193±0,0047 | 764 |
| 50 | -59,09 | 7,45 | -0,0078±0,0055 | 807 |
| 70 | -58,30 | 6,46 | 0,0023±0,0015 | 826 |
| 100 | -56,97 | 6,00 | -0,0004±0,0067 | 859 |
| 150 | -55,52 | 6,46 | -0,0100±0,0068 | 849 |
| 200 | -56,56 | 7,03 | 0,0094±0,0031 | 859 |
| 250 | -54,03 | 5,57 | -0,0580±0,0069 | 863 |
| 300 | -47,63 | 5.91 | -0,0657±0,0061 | 863 |
| 400 | -33,62 | 7,11 | -0,0366±0,0049 | 868 |
| 500 | -22,22 | 7,45 | -0,0078±0,0047 | 868 |
| 700 | -6,79 | 7,30 | -0,0071±0,0025 | 874 |
| 850 | 0,76 | 7,78 | 0,0086±0,0045 | 881 |
| 925 | 3,92 | 9,00 | 0,0161±0,0039 | 879 |
| 1000 | 2,62 | 8,71 | 0,0124±0,0098 | 170 |

**Table.1.** The results of determining the temperature coefficient for each isobaric surface.