# Peer review of "ACCOUNTING FOR METEOROLOGICAL EFFECTS IN THE DETECTOR OF THE CHARGED COMPONENT OF COSMIC RAYS"

_Geoscientific Instrumentation, Methods and Data Systems, 2021_

## Author Response (AR1)

**Thank you very much for your comments and suggestions! We are sure that these comments will improve the quality of the paper and allow us to optimize our work**.

**Response to Anonymous Referee #1**

**Reviewer #1:**

1. One general comment is that the authors perform consequent corrections (first for pressure, then for temperature/height) but this does not guarantee the optimal solution, because they may correlate with each other. A more robust method can be a multi-regression analysis when both meteorological variables (barometric pressure and temperature/height) are considered simultaneously. The authors may want to comment on that.

**Response**

Thank you very much for your comment and suggestion. We didn't try multi-regression analysis yet. It will be the subject for the next publications. For the current purposes independent correlation coefficients $R^2$ 0,81 (pressure) and 0,89 (temperature) are enough.

Other comments:

Thank you very much for your comment and suggestion. We took into account your comments in the text and marked corrections with a red color.

1. Line 14: check coordinates: Moscow is in the northern and eastern hemispheres.

   Done.

2. Line 16: "temperature coefficients" -> "temperature correction coefficients".

   Done.

3. Line 18: please define what is the "upper air". Normally, the stratosphere is not considered as the upper atmosphere.

   We have added the reference to Kochin et al., 2021, where the Upper-Air Soundings is described (line 134).

4. Line 33-34: should "any of" be replaced by "both", since it is about coincidence?

   Done.

5. Lines 46 and 48: please check the coordinates, the hemispheres are wrong.

Corrected

**Response**

Thank you very much for your comment and suggestion.

6. Line 56: what is meant by the "nuclear-active" particles?

**Response**

Thank you very much for your comment and suggestion. We checked. This term is correct in English-speaking literature. For example:

https://iopscience.iop.org/article/10.1088/0305-4470/7/6/010/pdf

We added comments in the text: «(protons, neutrons and also charged pions и kaons)»

7. Line 57-58: the second part of the sentence "therefore, it is necessary" is not logically connected to the first part. Please revise.

**Response**

Thank you very much for your comment and suggestion. We've corrected this sentence:

"Muons are not nuclear-active particles (such as protons, neutrons, and also charged pions и kaons) and lose energy for the excitation and ionization of air atoms. Ionization losses depend on the amount of matter above the detector, therefore, the barometric effect must be taken into account. The altitude of muon generation in the $\pi$ / K-decays is temperature dependent, therefore the temperature effect in the atmosphere must be taken into account. (Dorman, 1972, 2004, 2006)".

8. (1). A standard way of barometric correction is via an exponential formula, not linear. The latter is an approximation working only in a narrow range of pressure changes. Please explain that the linear relationship is sufficient.

**Response**

Thank you very much for your comment and suggestion. You are absolutely right. This linear fit is a standard technique for determining the barometric coefficient of ground installations. Its correctness (linear) is shown in Fig. 2. Coefficient of determination is 0.89

9. Line 70: is CARPET-MOSCOW the same as Dolgoprudny?

**Response**

Thank you very much for your comment and suggestion. Yes. We added in the text additional comment: «(which is located at the Dolgoprudny Scientific Station of the Lebedev Physical Institute RAS, Moscow region)»

10. Line 74: why was only one month chosen for the analysis? Was the temperature stable?

**Response**

Thank you very much for your comment and suggestion. This month was chosen as the most stable in terms of variations in primary cosmic rays and temperature.

11. Line 79: How exactly were the NM data used?

**Response**

Thank you very much for your comment and suggestion. We've corrected this phrase:

«To prove that secondary CRs variations, associated with barometric effect are more significant than variations of primary CRs variations, we use pressure corrected data of the Moscow neutron monitor.»

12. Line 83: Fig.3 contains no "upper horizontal time series" whatever it could be.

**Response**

Thank you very much for your comment and suggestion. We've corrected this phrase: «Fig. 3 shows neutron monitor count rate variations on the data of 2019. The black horizontal line is the average count rate [pulses/min] according to the annual data. Black vertical dashed lines are the boundaries of the months. The names of the month are signed at the bottom. The standard deviations for the data of each month are shown at the top.»

13. Line 88: what is \beta_{\sigma p}? Is it the same as \beta in line75?

**Response**

Thank you very much for your comment and suggestion. No, it's multiplication. We've correct it in the text: «$\beta \cdot \sigma_P = 0.018$ (1.8%),».

14. Line 95-98: the positive and negative effects need to be explained in more detail as a reader may not be familiar with that.

**Response**

Thank you very much for your comment and suggestion. We've corrected this phrase: «The temperature effect has two components: negative and positive. The negative temperature effect is associated with a decrease in muon fluxes during heating and expansion of the atmosphere. The positive temperature effect is associated with the appearance of additional muons, due to a decrease in the density of the atmosphere and, in connection with this, a decrease in the probability of interaction of charged pions and kaons with air nuclei. As a consequence, the probability of decays of charged pions and kaons and the appearance of additional muons increases. These two effects (positive and negative) are competitive»

15. 3: the \Delta H-effect is also usually modeled by an exponential relationship.

**Response**

Thank you very much for your comment and suggestion. You are absolutely right. This linear fit is a standard technique for determining the temperature coefficient of ground installations. Its correctness (linear) is shown in Fig. 6. Coefficient of determination 0.81

16. Line 156: what is the "temperature coefficient density"? The density of what?

**Response**

Thank you very much for your comment and suggestion. Corrected. «density of the temperature coefficient»

17. 4: panels d and e, representing different correction methods, are significantly different from each other, suggesting that the correction methods do not agree. The authors should comment on that and propose a preferred method.

**Response**

Thank you very much for your comment and suggestion. We added additional conclusions:

«This paper describes the CARPET installation, designed for detecting the charged component of secondary CRs. The barometric coefficient was determined using the built-in pressure sensor. The temperature coefficient was determined by two methods using the data of the upper-air sounding. The integral method for determining the temperature effect is the most accurate, however, due to the lack of regular measurements at high altitudes (since not all sounds reach high altitudes), it can be seen that the data processed by this method are less accurate. In this connection, it is more optimal to use the method of the effective generation level, since it does not require a complete temperature profile. Also, for the CARPET-MOSCOW installation, it is possible to use only the negative component of the temperature effect, since variations of the count rate have good ($R^2 = 0.8191$) correlation with $\Delta H$.»

18. Table 1: what is the last column (n)?

**Response**

Thank you very much for your comment and suggestion. We've corrected it in the text: «The results are shown in Table 1: the first column is the atmospheric pressure on the given surface, the second

column is the average temperature according to the data for 2019 - 2020, the third column is the standard deviation of the temperature, the fourth column is the temperature coefficient for the given isobaric surface, the fifth column is number of measurements (number of launches at which the sound reached the required altitude).»

**Response to Anonymous Referee #2**

page 1, lines 14f:

The geographic coordinates of the Dolgoprudny station are not correct. Moscow is not in the South hemisphere and not in the West of the prime meridian (Greenwich). Please check. In addition, the meaning of the parameter "Rc" should be given in the text.

**Response**

Thank you very much for your comment and suggestions. We've corrected it in the text.

The meaning of the parameter Rc is given in the Adstract. We have added it on line 50 in the text.

page 1, line 16:

"... barometric and temperature coefficients ..." -->

"... barometric and temperature correction coefficients ..."

page 1, line 18:

Please give information about what exactly you mean with "upper-air sounding of the atmosphere".

We have added the reference to Kochin et al., 2021, where the Upper-Air Soundings is described (line 134).

page 2, line 36:

I recommend to write "atmospheric pressure" instead of only "pressure".

**Response**

Thank you very much for your comment and suggestion. We've corrected it in the text.

page 2, lines 37ff:

"The CARPET installation detects particles of the following energies: in the UP and the LOW channels there are electrons and positrons with energies E> 200 keV, protons with E> 5 MeV, muons with E> 1.5 MeV, and photons with E> 20 keV (efficiency <1%).": Is "(efficiency <1%)" valid for all particle types or only for photons?

**Response**

Thank you very much for your comment and suggestion. Gas-discharge counters have some sensitivity to photons, mainly due to the Compton effect on the shells of the counter.

The G-M counters have ~100% response to the charged particles. Added in the text, line 40.

page 2, lines 46ff:

Please check the coordinates of the stations in this paragraph and if the effective cutoff rigidity is computed for the correct location. In addition, I would appreciate if you could give the time epoch for which Rc is given.

**Response**

Thank you very much for your comment and suggestion. We've corrected it in the text.

page 2, first paragraph under "2.1 Barometric effect":

I recommend to move the first paragraph starting with "Ground-based CARPET installations detect secondary charged particles, ..." under "2. Instrumentation and data analysis".

**Response**

Thank you very much for your comment and suggestion. We've corrected it in the text.

page 2, line 56:

What do you understand under "nuclear-active particles"?

Thank you very much for your comment and suggestion. We checked. This term is correct in English-speaking literature. For example:

https://iopscience.iop.org/article/10.1088/0305-4470/7/6/010/pdf

We added comments in the text: «(protons, neutrons and also charged pions и kaons)»

page 2, line 70:

The meaning of $\sigma_N$ and $\sigma_p$ should be given in the text.

**Response**

Thank you very much for your comment and suggestion. We've corrected it in the text:

«According to the data for 2019, hourly averaged average count rate and atmospheric pressure for the CARPET-MOSCOW installation $N_0 = 53667$ pulses/h, mean square deviation of the count rate $\sigma_N = 2187$ pulses/h; $P_0 = 988.7$ hPa, mean square deviation of the atmospheric pressure $\sigma_P = 9.8$ hPa.»

page 3, lines 74f:

You write that you selected June 2019 for the determination of the barometric coefficient $\beta$ as during this month there were no large geomagnetic and solar disturbances. From Figure 3 I would expect that the months July and August may be even more appropriate. To disentangle the barometric effect from the temperature effect, I would expect that it makes sense to also use the temperature in the atmosphere as a criterion for the selection of the time interval for the determination of the barometric coefficient.

**Response**

Thank you very much for your comment and suggestion. There were no substantial differences in the atmospheric parameters in July and August compared to June. We've corrected it in the text:

«For calculating the barometric coefficient $\beta$, it is necessary to determine the linear relationship between $\frac{\Delta N}{N_0}$ and $\Delta P$ (Fig. 2). Barometric coefficient $\beta$ for the CARPET-MOSCOW (which is located at the Dolgoprudny Scientific Station of the Lebedev Physical Institute RAS, Moscow region) installation is determined on the data of June 2019 (During this period there were no significant geomagnetic, solar and temperature disturbances): $\beta = -0.1861 \pm 0.0025\%$/hPa; coefficient of determination $R^2 = 0.8975$. Using Eq. (1), we obtain pressure-corrected data:»

page 3, line 80:

I would change: "Average count rate" to "Average pressure corrected count rate".

**Response**

Thank you very much for your comment and suggestion. We've corrected it in the text.

page 3, line 88:

Definition of $\beta \sigma_p$?

**Response**

Thank you very much for your comment and suggestion. It's multiplication. We've correct it in the test: «$\beta \cdot \sigma_P = 0.018\ (1.8\%)$,».

page 3, "2.2 Temperature effect":

I would appreciate if you could give a short description of the physics behind the temperature effect.

**Response**

Thank you very much for your comment and suggestion. We've corrected this phrase: «The temperature effect has two components: negative and positive. The negative temperature effect is associated with a decrease in muon fluxes during heating and expansion of the atmosphere. The positive temperature effect is associated with the appearance of additional muons, due to a decrease in the density of the atmosphere and, in connection with this, a decrease in the probability of interaction of charged pions and kaons with air nuclei. As a consequence, the probability of decays of charged pions and kaons and the appearance of additional muons increases. These two effects (positive and negative) are competitive»

page 3, lines 92ff:

I would write here something like:

"The muon component of secondary CRs is characterized by a significant temperature effect (Yanke, et al., 2011). To correct the CR measurements for this effect, it is necessary to carry out temperature measurements in the atmosphere close to the location of the CR instrument."

**Response**

Thank you very much for your comment and suggestion. We've corrected this phrase as you offered.

page 5, line 146:

Can you comment on the quality of the fit with $R^2 = 0,0049$.

**Response**

Thank you very much for your comment and suggestion.

The positive temperature effect has the greatest impact on high energy particle detectors. In this case, we have shown that there is practically no positive effect for this low-energy detector.

page 5, line 147:

"As seen in Fig. 7, there is a slight positive temperature effect.": Can you give here some quantitative information. From comparing Fig. 4 c) with Fig. 4 d) it is hard to see any differences between the two curves.

**Response**

Thank you very much for your comment and suggestion.

The positive temperature coefficient is given on line 162:

$\alpha_T = 0.0080 \pm 0{,}0038\%/°C$; coefficient of determination $R^2 = 0{,}0049$.

page 6, line 156:

What is exactly the "temperature coefficient density"? Units of $\alpha$?

**Response**
Thank you very much for your comment and suggestion.

$\alpha(x)$ –density of the temperature coefficient [%·°C$^{-1}$·hPa$^{-1}$], $\alpha(P)$ – temperature coefficient for a given isobaric surface [%/°C].

page 6, formula (7):

According to formula (6), I would expect also a $\Delta x$ in formula (7).

**Response**
Thank you very much for your comment and suggestion. Corrected.

page 6, line 171:

This formula has again the number (7) as above before line 165.

**Response**

Thank you very much for your comment and suggestion. Corrected.

page 7, "3. Conclusion":

"The results obtained by the effective generation method and the integral method correlate with each other.": What do you mean with "correlate with each other"? From Fig. 4 graphs c) and e), it seems that they show quite large differences. Can you give here some quantitative information? E.g. ratio curve c) vs. curve e). It seems that the curve in Fig. 4 e) shows more pronounced variations than the curve in Fig. 4 c). See e.g. the decrease after

day 10. Which of the curves c) or e) of Fig. 4 correspond better to the CR intensity near Earth? E.g. comparison with neutron monitor data which show almost no temperature effect.

**Response**

Thank you very much for your comment and suggestion. We've added to the text:

«There is also a correlation between the data of the Moscow neutron monitor (data corrected for the barometric effect) and CARPET-MOSCOW. According to the data for 2019-2020, for the initial CARPET-MOSCOW data: the correlation coefficient $R = 0.34$, for the CARPET-MOSCOW data corrected for the barometric effect: $R = 0.36$, for the CARPET-MOSCOW data corrected for the barometric effect and negative temperature effect: $R = 0.38$, for the CARPET-MOSCOW data corrected for the negative and the positive temperature effect and barometric effect: $R = 0.39$, for the CARPET-MOSCOW data corrected for the temperature effect according to the integral method and the barometric effect: $R = 0.2$.

"Comparison of Figures 4c and 4d shows that the contribution of the positive temperature effect is small. Comparison of Figures 4d and 4e demonstrates that the efficiency of data correction using the integral method is worse than using the effective generation method.

We can compare the efficiency of the correction for positive and negative temperature effects by comparing the CARPET-MOSCOW data with the data of a neutron monitor, which is practically not sensitive to the influence of temperature. The correlation coefficient between the pressure-corrected neutron monitor data for the period of 2019-2020 and the CARPET-MOSCOW data corrected for pressure and the negative temperature effect is $R = 0.38$, while taking into account the positive temperature effect is $R = 0.39$. Thus, the contribution of the correction for the positive temperature effect to the results of the CARPET-MOSCOW installation is small".

page 7, "4. Acknowledgments":

In addition to NMDB you should thank the IZMIRAN group (operator of neutron monitor station Moscow). Should you also acknowledge the Federal State Budgetary Institution «Central Aerological Observatory» (CAO)?

**Response**

Thank you very much for your comment and suggestion. We've added to the text:

«The authors express their gratitude to the Neutron Monitor Database (NMDB) team (www01.nmdb.eu) and IZMIRAN team (https://www.izmiran.ru/) for the data from the ground network of neutron monitors and Federal State Budgetary Institution «Central Aerological Observatory» (CAO) team (http://www.cao-rhms.ru/) for providing the data of upper-air sounding of the atmosphere for 2019-2020.»

page 11, Fig. 3:

Does Fig. 3 show "measured count rates" or "pressure corrected measured count rates"?

**Response**

Thank you very much for your comment and suggestion. We've added to the text:

«**Fig. 3.** Pressure corrected count rate variations of the Moscow neutron monitor for the period of 2019. Horizontal line – average count rate.»

page 12, Fig. 4:

Does Fig. 4 show data for the years 2019/2020? What is shown by the grey curve and what by the black curve?

**Response**

Thank you very much for your comment and suggestion. We've added to the text:

**«Fig. 4** Count rate variations of the CARPET-MOSCOW installation for the period of 2020-2021: $a$ – uncorrected data, $b$ – pressure corrected data, $c$ - pressure and temperature (negative effect applying the effective generation method) corrected data, d – pressure and temperature (negative and positive effect applying the effective generation method) corrected data, e – pressure and temperature (the integral method) corrected data. Grey lines - initial data, Black lines – data with averaging by 24 points.»